# Integrated Analysis of Transcriptomic and Metabolomic Data Reveals the Mechanism by Which LED Light Irradiation Extends the Postharvest Quality of Pak-choi (*Brassica campestris* L. ssp. *chinensis* (L.) Makino var. *communis* Tsen et Lee)

**DOI:** 10.3390/biom10020252

**Published:** 2020-02-07

**Authors:** Zhicheng Yan, Jinhua Zuo, Fuhui Zhou, Junyan Shi, Dongying Xu, Wenzhong Hu, Aili Jiang, Yao Liu, Qing Wang

**Affiliations:** 1Key Laboratory of the Vegetable Postharvest Treatment of Ministry of Agriculture, Beijing Key Laboratory of Fruits and Vegetable Storage and Processing, Key Laboratory of Biology and Genetic Improvement of Horticultural Crops (North China) of Ministry of Agriculture, Key Laboratory of Urban Agriculture (North) of Ministry of Agriculture, National Engineering Research Center for Vegetables, Beijing Academy of Agriculture and Forestry Sciences, Beijing 100097, China; 18840821461@163.com (Z.Y.); zuojinhua@126.com (J.Z.); zhoufuhuihui@126.com (F.Z.); shijunyan0130@126.com (J.S.); happyyaoyaoer@126.com (Y.L.); 2Laboratory of Biotechnology and Bioresources Utilizatio, Ministry of Education, College of Life Science, Dalian Minzu University, Dalian 116600, China; 18840875773@163.com (D.X.); hwz@dlnu.edu.cn (W.H.); jal@dlnu.edu.cn (A.J.)

**Keywords:** Pak-choi, LED irradiation, RNA-seq, metabolomics, postharvest quality

## Abstract

Low-intensity (10 μmol m^−2^ s^−1^) white LED (light-emitting diode) light effectively delayed senescence and maintained the quality of postharvest pakchoi during storage at 20 °C. To investigate the mechanism of LED treatment in maintaining the quality of pakchoi, metabolite profiles reported previously were complemented by transcriptomic profiling to provide greater information. A total of 7761 differentially expressed genes (DEGs) were identified in response to the LED irradiation of pak-choi during postharvest storage. Several pathways were markedly induced by LED irradiation, with photosynthesis being the most notable. More specifically, porphyrin and chlorophyll metabolism and glucosinolate biosynthesis were significantly induced by LED irradiation, which is consistent with metabolomics reported previously. Additionally, chlorophyllide a, chlorophyll, as well as total glucosinolate content was positively induced by LED irradiation. Overall, LED irradiation delayed the senescence of postharvest pak-choi mainly by activating photosynthesis, inducting glucosinolate biosynthesis, and inhibiting the down-regulation of porphyrin and chlorophyll metabolism pathways. The present study provides new insights into the effect and the underlying mechanism of LED irradiation on delaying the senescence of pak-choi. LED irradiation represents a useful approach for extending the shelf life of pak-choi.

## 1. Introduction

Pak-choi (*Brassica campestris* L. ssp. *Chinensis*) is a cruciferous vegetable native to China. Currently, pak-choi has also become more popular and widely used in Western diets [1]. Pak-choi contains high levels of several nutrients that are beneficial to human health, including ascorbic acid, glucosinolates, and polyphenols [2,3]. A high level of consumption of cruciferous vegetables has been related with a reduced risk of lung and colorectal cancer [4] and epidemiological studies have provided evidence that the consumption of cruciferous vegetables more effectively protects against cancer than the total intake of mixed fruits and vegetables [5]. Pak-choi, however, suffers from several postharvest problems that affect its sensory and nutritional quality, including weight loss, wilting, and leaf yellowing. Thus, several approaches have been explored to maintain the postharvest quality of pak-choi, such as the application of aqueous chlorine dioxide combined with ultraviolet-C irradiation [6], and low temperature storage [7]. Greater efforts are needed, however, to identify a cost-effective, reliable approach to extending the postharvest quality of harvested produce.

Light is one of the most important environmental variables affecting the concentrations of phytochemicals in plants [8]. In this regard, the use of light emitting-diodes (LED) represents an efficient and economical approach to administer light treatments to vegetables because of their small size, safety, stability, and cold light illuminator [9,10]. The use of LED lighting for the postharvest treatment of many different vegetables has been reported. Exposure to white–blue LED light delayed the senescence of harvested broccoli [11], red LED light suppressed the postharvest yellowing of broccoli and reduced the loss of ascorbic acid [12], LED White light, red light irradiation can improve the nutrition and flavor quality of post-harvest citrus [13], and LED light treatment can induce the ripening of bananas and improve their quality and nutrition potential [14]. Thus, it appears that LED irradiation may have broad applications for the postharvest preservation of vegetables. Additional research is required, however, to determine the mechanism by which LED light exposure maintains the postharvest quality of vegetables in order to optimize its use.

Our previous study demonstrated that 10 μmol m^−2^ s^−1^ white LED light irradiation maintained the quality, increased the antioxidant capacity, and regulated the chlorophyll metabolism of postharvest pakchoi [15]. Additionally, we used an untargeted metabolomic approach to investigate the effect of LED irradiation on the metabolite profile of pak-choi, the data denoted that LED irradiation kept the quality of pak-choi by regulating several metabolic processes, elevating folate, thiamine, glutathione, riboflavin, and total carotenoid content, decreasing the degradation of glucosinolate, and enhancing key synthetic precursors of chlorophyll [16].

Transcriptomic and metabolomic analyses are useful approaches for identifying gene-to-metabolite networks involved in the physiological responses of plants [17,18]. The integration of differentially-expressed gene profiles with metabolite data is an effective way to better understand gene-to-metabolite networks [19], the function of differentially-expressed genes [20], and to integrate biological information and broaden our knowledge of plant metabolism [21]. The integrated use of transcriptomics and metabolomics has been used to investigate the metabolic changes that occur during the postharvest storage of produce. For example, Ding et al. [22] identified a network of processes that occur during the postharvest senescence process in citrus fruits. Thus, in the present study, we augmented transcriptomic profiling and performed a joint analysis with previous metabolomics to improve understanding of the response of harvested pak-choi to LED irradiation.

## 2. Material and Methods

### 2.1. Plant Material and LED Treatment

Sample treatment and LED parameter was as previously described [15]. Pak-choi (*Brassica campestris* L. ssp. *Chinensis*) plants were harvested in Xiaotangshang, Beijing (China) at their optimum harvest date and immediately transported to the laboratory. Undamaged pak-choi plants with thick, fleshy, firm stalks and glossy, dark-green leaves with no evidence of mechanical damage were selected to study. The selected plants were then randomized and partitioned into two groups, with each group containing approximately 2 kg of plants. Subsequently, each harvested pakchoi plant was placed in a polyethylene bag (0.03 mm) and stored at 20 ± 1 °C at approximately 90% relative humidity. The control group was stored under dark conditions, while the LED-treated group was stored under continuous irradiation with white LED light at wavelengths of 448 nm and 549 nm at a light intensity of approximately 10 μmol m^−2^ s^−1^. Other parameters of white LED light were red (600–700 nm) at 22.221 μmol m^−2^ s^−1^, green (500–599 nm) at 43.325 μmol m^−2^ s^−1^, blue (400–499 nm) at 25.461 μmol m^−2^ s^−1^, far-red (701–780 nm) at 2.3373 μmol m^−2^ s^−1^, and PPF (400–700 nm) at 91.0 μmol m^−2^ s^−1^. The red/far-red ratio was 9.51 and the red/blue ratio was 0.73. The light intensity was measured using a Lighting Passport (ALP-01, Asensetek, Taiwan). Approximately 250 g of pak-choi were collected from each group on each sample day and mixed to obtain a homogeneous sample. The liquid nitrogen was used to freeze the collected material, and stored the collected material at –80 °C until subsequent analysis. Our previous study indicated that pak-choi stored in the dark (control) conditions exhibited a significant decrease in quality after 5 d. Therefore, three biological replicates were collected just prior to treatment (T01, T02, T03), and three biological replicates of pak-choi stored in the dark (T04, T05, T06) or under LED lights (T07, T08, T09) were collected at 5 d and subjected to transcriptomic analyse.

### 2.2. RNA-Seq Library Preparation and Sequencing

The RNA-seq library preparation and sequencing were determined using the method depicted by [23]. RNA concentration was measured using NanoDrop 2000 (Thermo Fisher Scientific, Waltham, MA, USA). RNA integrity was assessed using the RNA Nano 6000 Assay Kit of the Agilent Bioanalyzer 2100 system (Agilent Technologies, Santa Clara, CA, USA). A total amount of 1 μg RNA per sample was used as input material for the RNA sample preparations. Sequencing libraries were generated using NEBNext UltraTM RNA Library Prep Kit for Illumina (NEB, Ipswich, MA, USA) following the manufacturer’s recommendations and index codes were added to attribute sequences to each sample. 

### 2.3. Transcriptome Data Analysis

#### 2.3.1. Quality Control

Raw data (raw reads) of fastq format were firstly processed through in-house perl scripts. In this step, clean data (clean reads) were obtained by removing reads containing adapter, reads containing ploy-N and low-quality reads from raw data. At the same time, Q20, Q30, GC-content and sequence duplication level of the clean data were calculated. All the downstream analyses were based on clean data with high quality.

#### 2.3.2. Comparative Analysis

The adaptor sequences and low-quality sequence reads were removed from the data sets. Raw sequences were transformed into clean reads after data processing. These clean reads were then mapped to the reference genome sequence. Only reads with a perfect match or one mismatch were further analyzed and annotated based on the reference genome. Hisat2 tools soft were used to map with reference genome.

#### 2.3.3. Functional Annotation of Genes

Genes were identified and annotated using the following databases: Nr (ftp://ftp.ncbi.nih.gov/blast/db/FASTA/); Pfam (http://pfam.xfam.org/); KOG/COG (http://www.ncbi.nlm.nih.gov/KOG); Swiss-Prot (http://www.uniprot.org/); KEGG (http://www.genome.jp/kegg/); and GO (http://www.geneontology.org/). 

#### 2.3.4. Quantification of Gene Expression Levels

Gene expression levels were estimated using fragments per kilobase of transcript per million fragments mapped (FPKM) based on the following formula: (1)$$\mathrm{FPKM}=\frac{\mathrm{cDNA}\;\mathrm{Fragments}}{\mathrm{Mapped}\;\mathrm{Fragments}\;\left(\mathrm{Millions}\right)\times \mathrm{Transcript}\;\mathrm{Length}\;\left(\mathrm{kb}\right)}$$

#### 2.3.5. Differential Expression Analysis

Differential expression analysis of two conditions/groups was performed using the DESeq R package (1.10.1, http://www.bioconductor.org/packages/release/bioc/html/DESeq.html). DESeq [24] provides statistical routines for determining differential expression in gene expression data using a model based on the negative binomial distribution. Expression values were adjusted to control the false discovery rate as described by [25]. Differences in the expression of genes having an adjusted *p*-value < 0.05 were designated as differentially expressed genes (DEGs). The FDR (false discovery rate) was adjusted by PPDE (posterior probability of being DE). The threshold for significantly differential expression was set as FDR < 0.01 & |log2 (foldchange)| ≥2.

### 2.4. Sample Preparation and Metabolomic Analysis

Sample preparation and the assay procedure used followed previous experiments [16,26].

### 2.5. Transcriptome and Metabolome Coalition Analysis

Coalition of KEGG pathway enrichment of DEGs and differentially expressed metabolites was conducted using data on differentially expressed genes and metabolites that were enriched in the same pathway as determined by the KEGG analysis. 

#### 2.5.1. Correlation Analysis

Pearson correlation coefficients (PCC) and P-values were calculated separately between log_2_(genes) and log_2_(metabolites). Genes and metabolites were selected following the standards of PCC > 0.8 and *p*-value < 0.05.

#### 2.5.2. Correlation Network Visualization

A network model was constructed to illustrate the relationship between genes and metabolites. Genes and metabolites enriched in the same pathways and with correlation coefficients greater than 0.8 were selected for use in the model.

#### 2.5.3. Canonical Correlation Analysis

Canonical correlation analysis (CCA) was used to explore the relationships between differentially-expressed genes and metabolites that were in the same KEGG enriched pathways. CCA was used to identify linear combinations between the two variables and identify maximum correlations. 

### 2.6. Glucosinolates

Total glucosinolate content was determined by the method depicted by [27]. Frozen pak-choi tissue powder (0.05 g) was blended with 2.8 mL of a solution of methanol-acetic acid (40% methanol and 0.5% acetic acid; avoiding total glucosinolate hydrolysis by endogenous myrosinase), and another equivalent of it was added to 2.8 mL deionized water (blank). The two mixtures were incubated at 37 °C for 15 min, adding 2.1 mL methanol and 3 mg of activated carbon to each sample to stop the reaction. The samples were centrifuged twice at 12,000× *g* for 15 min at 4 °C. One mL of the supernatant was then added to a 4 mL standard glucose solution and incubated at 37 °C for 30 min. Adding 4 mL H_2_SO_4_ to end the reaction, measured at 540 nm of absorbance. The content of glucose in the reaction solution was the standard to compute the content of total glucosinolate.

### 2.7. Chlorophyll

Chlorophyll content was determined using a modified version of the method reported by [28]. Pak-choi tissue powder (1 g) was homogenized in 10 mL acetone: ethanol (2:1), then the mixture was centrifuged at 12,000× *g* for 10 min at 4 °C. The total chlorophyll content was determined by measuring sample absorbance at 645 nm and 663 nm.

### 2.8. Chlorophyllide a

Pak-choi tissue (0.5 g) was extracted with 9 mL phosphate buffer (50 mM, pH 7.4) and the mixture was then centrifuged at 13,000× *g* at 4 °C for 15 min. The supernatant was then collected. Chlorophyllide a content was determined using an ELISA Kit (LMAI Bio, Shanghai, China) at 450 nm with a microplate reader (Multiskan GO, Thermo Scientific, Waltham, MA, USA) according to the manufacturer’s protocol. Standard was used to generate a standard curve under the same condition, and chlorophyllide a content was calculated based on the standard curve.

### 2.9. Statistical Analyses

All statistical were analyzed by SPSS 19.0 (SPSS Inc., Chicago, IL, USA). The one-way ANOVA was used on data, and means were compared by LSD test at a significance level of *p* < 0.05.

## 3. Results and Discussion

### 3.1. Transcriptome Response and Joint Analysis

A transcriptomic (Appendix A) was conducted and analyzed in conjunction with previous metabolomic [16] of pak-choi during storage to characterize the regulatory mechanism underlying the white-light LED delay of senescence. The three groups were Initial (prior to treatment), CK 5 d (pak-choi stored in the dark for 5 d) and LED 5 d (pak-choi irradiated with LED lights for 5 d). The correlation among the three sample-types was analyzed. The correlation analysis (Figure 1A) indicated a clear separation between the three groups of samples, demonstrating that the irradiation of pak-choi with LED white light alters gene expression. A total of 7761 DEGs [|log2(foldchange)| ≥ 2 (FC ≥ 2)] (Appendix A); false discovery rate ≤ 0.01 (FDR ≤ 0.01)] were identified between the three sample types (Initial, CK 5 d, and LED 5 d). This included 6692 DEGs (2733 up- and 3959 down-regulated) in the Initial/CK 5 d comparison, 3906 DEGs (1572 up- and 2334 down-regulated) in the Initial/LED 5 d comparison, and 1339 DEGs (1011 up and 328 down-regulated) in the CK 5 d/LED 5 d comparison (Figure 1B), A Venn diagram (Figure 1C) illustrates the distribution of DEGs among the three treatment groups.

Relative to the Initial/LED 5 d comparison, twice as many DEGs were identified in the Initial/CK 5 d comparison (Figure 1B). This indicates that the white LED light treatment inhibited many of the changes in gene expression that occurred during the postharvest storage of pak-choi under dark conditions. A heat map of gene expression (Appendix A) indicated that expression of most of the genes in the LED samples were down-regulated relative to the initial samples and only a few of genes were up-regulated. In most cases, the level of gene expression in the LED-treatment samples was between the CK 5 d (dark-stored) and initial samples, and more similar to the expression levels in the initial samples. This indicates that white LED light treatment inhibited many of the senescence-related changes in gene expression that happen in pak-choi during dark storage. The data indicate that the use of a white LED treatment may be an effective way to maintain the quality of pak-choi during storage by inhibiting senescence-related gene expression. LED irradiation has been shown to keep the quality of several vegetable crops after harvest. Ma et al. [12] reported the reduced loss of ascorbic acid (AsA) by exposure to white LED light was highly regulated at the transcriptional level. Hasperué et al. [11] demonstrated that continuous white–blue LED light can delay the senescence of broccoli after harvest. JoaquÃn et al. [29] reported that LED light can extend the postharvest quality of brussels sprouts during storage.

Kyoto Encyclopedia of Genes and Genomes (KEGG) enrichment analysis revealed that many metabolic-, amino acid-, and enzyme activity-related pathways were affected by the LED white light treatment (Figure 1D). Photosynthesis was the most obviously impacted. Previous metabolomic data indicated that porphyrin and chlorophyll metabolism and glucosinolate biosynthesis were stimulated by LED irradiation [16]. In this study, these two pathways were also identified by KEGG enrichment analysis. Thus, three representative pathways were selected for further analysis including photosynthesis, porphyrin and chlorophyll metabolism, and glucosinolate biosynthesis.

### 3.2. Photosynthesis

Photosynthesis involves two processes: light reactions that produce ATP and NADPH, and light-independent carbon reactions that fix atmospheric CO_2_ into organic molecules using the generated ATP and NADPH [30]. Transcriptome analysis indicated that the LED white light treatment significantly affected the light reactions (PLR) of photosynthesis (Figure 2A). Maintaining PLR would have a positive effect on pak-choi quality during postharvest storage because the availability of greater levels of photoassimilates could delay senescence [31]. Braidot et al. [32] reported that a light treatment of lamb’s lettuce activated photosynthesis and the maintenance of PLR promoted the production of ATP. A higher energy status can delay senescence [33], as ATP is involved in anti-oxidative processes [34]. Our transcriptomic data indicated that 49 genes (Appendix A) related to PLR were among the DEGs (FC ≥ 2; FDR ≤ 0.01) in the LED treatment samples. The identified genes were related with Photosystem II, Photosystem I, Cytochrome b6f complex, photosynthethic electron transport, and F-type ATPase in PLR. A contrast of the expression profiles of all the treatment sample types indicated that, relative to dark-stored (control) and LED treatment samples, the initial samples had the supreme extent of PLR-related gene expression. In contrast, all of the PLR-related genes exhibited down-regulation after storage for 5 days in both the dark-stored (control) and LED treatment samples. Compared to the dark-stored (control) samples, however, the extent of down-regulation was obviously less in the LED treatment samples (Figure 2B), indicating that that LED treatment helped to maintain PLR-related gene expression in pak-choi during storage. These results indicate that the expression of PLR-associated genes was activated or at least partially maintained in pak-choi during postharvest storage by the white LED light treatment. A similar result was obtained by Hasperué et al. [29] using LED light to maintain photosynthetic activity, the green color and extend the shelf life of brussels sprouts. Braidot et al. [32] also improved the quality of lamb’s lettuce during storage with the use of low-intensity lighting.

### 3.3. Chlorophyll Synthesis

Green color, resulting from high levels of chlorophyll, is a quality parameter strongly desired by consumers as it reflects freshness. Chlorophyll degradation occurs rapidly during postharvest storage of pak-choi, reflected as a yellowing of the produce, and is one of the most obvious indicators of senescence. Therefore, maintaining chlorophyll levels in pak-choi is a direct way to preserve sensory quality. Light, as the important environmental factors for plants [12], is widely-used to keep the postharvest quality of vegetables. Braidot et al. [32] reported that low intensity light treatment of lamb’s lamb’s lettuce (*Valerianella olitoria* [L.] Pollich) during postharvest storage resulted in higher levels of chlorophyll. Büchert et al. [35] demonstrated that light could effectively maintain the postharvest chlorophyll content of broccoli by inhibiting *BoPPH* expression (a chlorophyll-degradation gene), and Jin et al. [36] indicated that a green LED light treatment maintained chlorophyll levels in harvested broccoli florets. In this study, KEGG enrichment analysis revealed that the white LED light treatment had a significant positive effect on porphyrin and chlorophyll metabolism pathways (Figure 3A). Two factors related to chlorophyll synthesis are present in porphyrin and chlorophyll metabolism pathways: the composite of chlorophyll precursors and the constitution of chlorophyll a/b. The synthesis of chlorophyll precursors from L-glutamic acid to protoporphyrin IX is catalyzed by a series of enzymes [37]. In the present study, six genes (Appendix A) related to chlorophyll synthesis were among the DEGs (FC ≥ 2; FDR ≤ 0.01) in this experiment. The expression of two genes related to the key enzyme, HemA, were differentially expressed in the three sample groups of pak-choi. Heat map analysis revealed that the initial samples had the highest level of expression of the HemA-related genes, and that after five days of storage the expression of these genes was up-regulated in the LED treatment group relative to the dark-stored (control) treatment group (Figure 3B). These results indicate that the expression of HemA-related genes was down-regulated in the dark-stored (control) samples and that the white LED light treatment could delay the down-regulation of these genes. As HemA catalyzes the formation of L-Glutamate 1-semialdehyde from L-Glutamyl-tRNA [38], the up-regulated expression of HemA-related genes was positive to the composite of chlorophyll precursors. Regarding the synthesis of chlorophyll a/b, significantly differences (FC ≥ 2; FDR ≤ 0.01) were observed in the expression of genes encoding three enzymes (chlI, chlE, and por) and the level of two metabolites (divinylchlorophyllide a, chlorophyllide a) [16]. The enzymes chlI, chlE, and por play an important role in catalyzing the synthesis of chlorophyll [39], as divinyl chlorophyllide a and chlorophyllide a are major components of chlorophyll synthesis [40]. In the current study, genes encoding chlI, chlE, and por were again highest in the initial sample group. However, the LED treatment helped to maintain the expression level of these genes relative to the dark-stored (control) sample group. These data indicate chlI, chlE, and por gene expression are down-regulated during the senescence of pak-choi but that the LED treatment delayed their down-regulation. The metabolomic analysis revealed a high level of divinyl chlorophyllide a and chlorophyllide a in the initial sample group and that the level of these metabolites significantly decreased in the dark-stored treatment group after five days of storage. In contrast, the levels of these metabolites were maintained in the LED treatment group of pak-choi after five days of storage [16]. Overall, the results of this study indicated that the LED treatment inhibited the decrease in chlorophyll levels that occurs in pak-choi during dark storage by maintaining the expression of genes encoding the enzymes (HemA, chlI, chlE, por) and the levels of major substrates (divinyl chlorophyllide a and chlorophyllide a) of chlorophyll biosynthesis. This finding was validated by determining chlorophyllide a and chlorophyll levels in the various treatment groups utilized in the current study (Figure 3C,D).

### 3.4. Glucosinolate Biosynthesis

Glucosinolate belongs to a class of secondary sulphur-containing metabolites produced by crucifers [41]. The major glucosinolate biosynthetic precursors are the amino acids, methionine, tryptophan and phenylalanine, that are used to produce aliphatic, indolic, and aromatic glucosinolates, respectively [42] (Figure 4A). Glucosinolates have been reported to function as against pathogen and deterrent action against insects [43]. Mithen et al. [44] reported that glucosinolates were toxic to *Leptosphaeria maculans*, and Mari et al. [45] observed that glucosinolates inhibited the conidial germination of postharvest pathogens of fruit. Glucosinolates and their metabolites were proved that effective reduce the risk of cancer in humans [46]. More specifically, Steinbrecher et al. [47] indicated that glucosinolates play a protective role against prostate cancer development possibly through the induction of biotransformation enzymes. In this study, the effect of LED treatment of pak-choi on the regulation of glucosinolate biosynthesis was investigated. Transcriptomic data identified seven DEGS (FC ≥ 2; FDR ≤ 0.01) (Appendix A) related to glucosinolate biosynthesis in LED-treated pak-choi. The identified genes were related to six key enzymes in the biosynthesis of glucosinolate, including MAM1, CYP79F1, CYP83A1, CYP83B1, SUR1, UGT74B1. MAM1, CYP79F1, CYP83A1 are in the methionine pathway, CYP83B1 is in the chorismite pathway, and SUR1 and UGT74B1 are present in two different pathways. A decline in the level of all of the identified DEGs related to glucosinolate synthesis was observed in the dark-stored (control) treatment group after 5 d of storage. In contrast, 3 DEGS related to MAM1 and CYP79F1 were higher in the LED treatment group after 5 d of storage, relative to the dark-stored (control) sample group, and 4 DEGS related to CYP83A1, CYP83B1, SUR1, and UGT74B1, were higher than in the initial and dark-stored (control) treatment groups (Figure 4B). This indicates that the LED treatment up-regulated the expression genes related to these enzymes.

Metabolomic data indicated that levels of S-(Indolylmethylthiohydroximoyl)-L-cysteine (SLC) and glucoiberverin in the LED treatment group were higher than in the other two treatment groups, but that the level of indolylmethyl-desulfoglucosinolate (ID) was lower [16]. SLC, ID and glucoiberverin play an important role in glucosinolate biosynthesis. SLC is a type of thiohydroximate, ID is a major substrate for the synthesis of indolylmethyl-glucosinolate [48], and glucoiberverin is an aliphatic glucosinolate. The higher levels of SLC and glucoiberverin and lower levels of ID in LED-treated pak-choi indicates that glucosinolate biosynthesis in pak-choi was induced by LED irradiation. Overall, LED treatment promoted glucosinolate biosynthesis and increased the level of glucosinolate in pak-choi (Figure 4C). An increase in glucosinolate levels in microgreens and baby, leafy-green brassica crops by LED light has been previously reported [49].

## 4. Conclusions

The present study indicated that the effect of LED irradiation on delaying senescence in pak-choi is related to the activation of photosynthesis, the induction of glucosinolate biosynthesis, and an inhibition of the down-regulation of porphyrin and chlorophyll metabolism pathways (Figure 5). White LED light treatment helped to up-regulate the expression of genes related to photosynthesis, maintain chlorophyll synthesis, and glucosinolate biosynthesis. Photosynthesis provided the energy needed to support ATP synthesis, while the maintenance of chlorophyll levels delayed the loss in sensory quality that typically occurs during the storage of pak-choi. The maintenance or increase in glucosinolate biosynthesis provided additional microbial and insect resistance and added nutritional value of pak-choi. All of the above contributed to the maintenance of the postharvest quality of pak-choi during storage. The application of LED light technology could be used to extend the shelf-life and improve the postharvest quality of pak-choi.

## Figures and Tables

**Figure 1 biomolecules-10-00252-f001:**
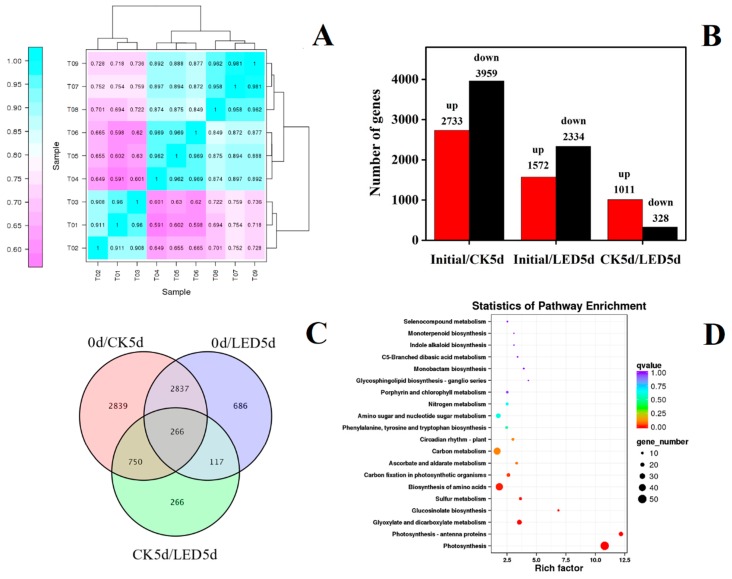
Expression and correlation analysis between pak-choi treatment groups. (**A**) Initial (T01, T02, T03), dark-stored (control) for 5 d (T04, T05, T06), and LED-irradiated for 5 d (T07, T08, T09). (**B**) The number of differentially expressed genes (DEGs) (|log2(foldchange)| ≥ 2 (FC ≥ 2); false discovery rate ≤ 0.01 (FDR ≤ 0.01)). (**C**) Venn diagram of DEGs (0d/CK5d, initial/dark-stored 5d; 0d/LED5d, initial/LED 5 d; CK5d/LED5d, dark-stored 5d/LED 5 d). (**D**) KEGG pathway enrichment analysis of DEGs.

**Figure 2 biomolecules-10-00252-f002:**
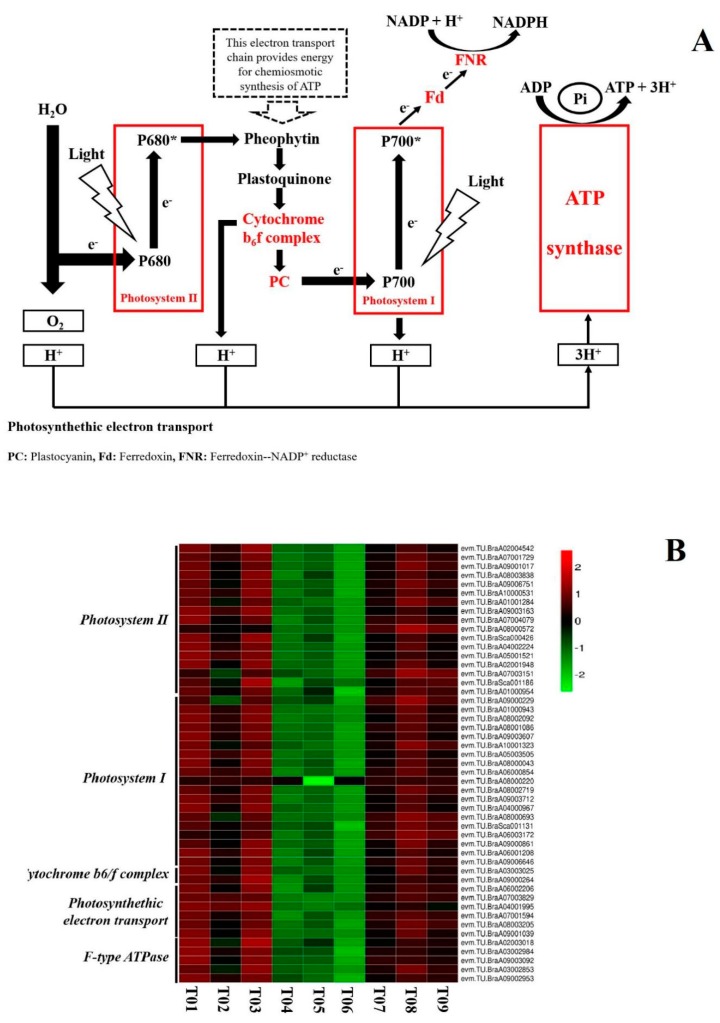
The effect of LED irradiation on the expression of genes involved in photosynthesis in pak-choi. (**A**) Metabolites or genes expressed at a higher level in the LED treatment group, relative to the dark-stored (control) treatment-group, are illustrated in red. Metabolites or genes expressed at a lower level in the LED treatment group, relative to the dark-stored (control) treatment group, are illustrated in green. (**B**) Expression pattern of genes involved in photosynthesis in response to LED irradiation. A total of 49 DEGs (FC ≥ 2; FDR ≤ 0.01) associated with photosynthesis were identified. Log2-based FPKM values were used to generate the heat map. The scale represents the relative signal intensity of FPKM values (T01, T02, T03: initial; T04, T05, T06: dark-stored control 5d; T07, T08, T09: LED 5 d).

**Figure 3 biomolecules-10-00252-f003:**
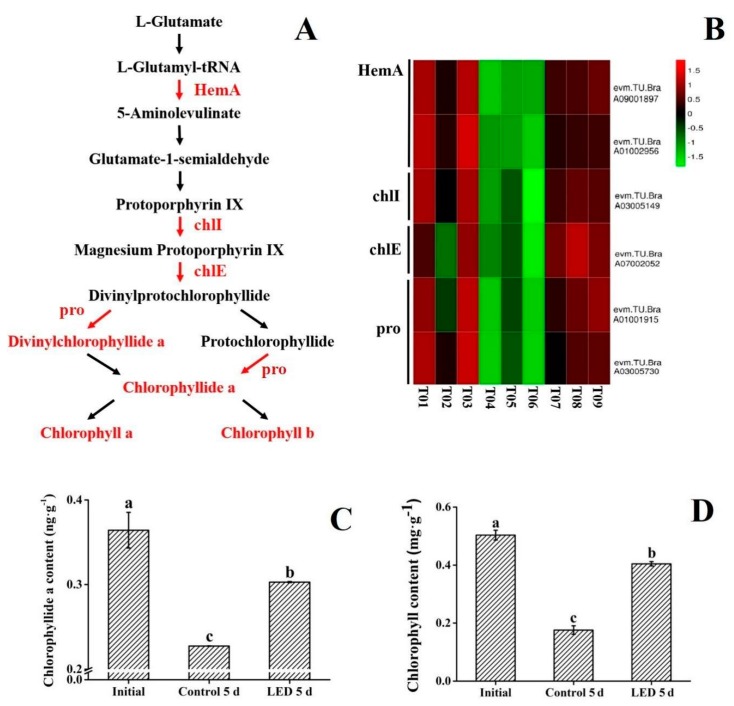
The effect of LED irradiation on the expression of genes involved in the chlorophyll synthesis and chlorophyll and chlorophyllide a content in pak-choi. (**A**) Metabolites or genes expressed at a higher level in the LED treatment group, relative to the dark-stored (control) treatment group, are illustrated in red. Metabolites or genes expressed at a lower level in the LED treatment group, relative to the dark-stored (control) treatment group, are illustrated in green. (**B**) Expression pattern of genes involved in the porphyrin and chlorophyll metabolism pathways in response to LED irradiation. Six DEGs (FC ≥ 2; FDR ≤ 0.01) associated with photosynthesis were identified. Log2 based FPKM values were used to create the heat map. The scale represents the relative signal intensity of FPKM values (T01, T02, T03: initial; T04, T05, T06: dark-stored control 5d; T07, T08, T09: LED 5 d). (**C**,**D**) Levels of chlorophyllide a (**C**) and chlorophyll (**D**) in the three treatment groups (initial, dark-stored control 5d, and LED treatment 5d). Vertical bars represent the standard error of the mean (*n* = 3). Means denoted by the same letter do not differ significantly at *p* < 0.05 as determined by LSD’s multiple range test.

**Figure 4 biomolecules-10-00252-f004:**
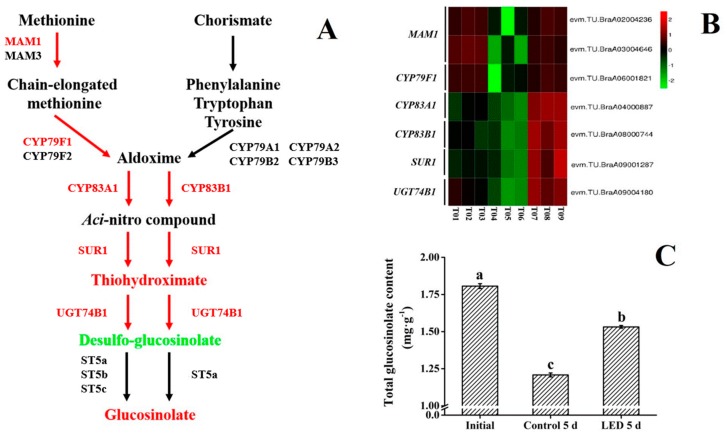
The effect of LED irradiation on the expression of genes in pak-choi involved in the biosynthesis of glucosinolate and levels of total glucosinolate. (**A**) Metabolites or genes expressed at a higher level in the LED treatment group, relative to the dark-stored (control) treatment group, are illustrated in red. Metabolites or genes expressed at a lower level in the LED treatment group, relative to the dark-stored (control) treatment group, are illustrated in green. (**B**) Expression pattern of genes involved in glucosinolate biosynthesis in response to LED irradiation. Seven DEGs (FC ≥ 2; FDR ≤ 0.01) associated with photosynthesis were identified. Log2 based FPKM value were used to generate the heat map. The scale represents the relative signal intensity of FPKM values (T01, T02, T03: initial; T04, T05, T06: dark-stored control 5d; T07, T08, T09: LED treatment 5d). (**C**) Level of total glucosinolate in pak-choi in the three treatment groups (initial, dark-stored control 5d, and LED treatment 5d) of pak-choi. Vertical bars represent the standard error of the mean (*n* = 3). Means denoted by the same letter do not differ significantly at *p* < 0.05 as determined by LSD’s multiple range test.

**Figure 5 biomolecules-10-00252-f005:**
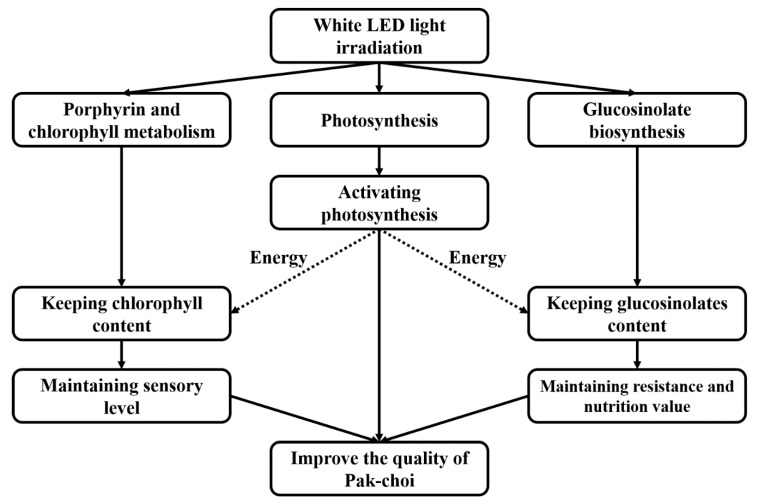
Diagrammatic model of the proposed mechanism responsible for the effect of LED irradiation on the maintenance of postharvest quality of pak-choi. White LED light treatment induces photosynthesis, chlorophyll synthesis, and glucosinolate biosynthesis. Photosynthesis provides the energy needed for ATP synthesis. The maintenance of chlorophyll synthesis during storage maintained the postharvest sensory quality of pak-choi. Increased levels of glucosinolate synthesis increased the level of microbial and insect resistance, as well as the nutritional value of pak-choi. Maintenance or increased levels of the above components extended and improved the postharvest quality of pak-choi during storage.

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
