# Peer review of "Integrated Analysis of Transcriptomic and Metabolomic Data Reveals the Mechanism by Which LED Light Irradiation Extends the Postharvest Quality of Pak-choi (Brassica campestris L. ssp. chinensis (L.) Makino var. communis Tsen et Lee)"

_biomolecules, 2020, doi:10.3390/biom10020252_

Round 1

Reviewer 1 Report

The manuscript entitled " Integrated analysis of transcriptomic and metabolomic data reveals the mechanism by which LED light irradiation extends the postharvest quality of pak-choi (Brassica campestris L. ssp. chinensis (L.) Makino var. communis Tsen et Lee)” concerns a study on the trascriptomic changes induced in pak-choi by LED treatments compared to control plants kept in the dark. In addition, the authors performed the analysis of specific metabolites (glucosinolates, chlorophylls and chlorophyllide a). The study is of interest because it offers several evidences of the metabolic pathways activated by LED treatments and their potential benefits in increasing post-harvesting time. I think that the paper is suitable for this journal after a careful revision of some minor points highlighted in the attached PDF file. In particular, it is important to remove the word “metabolomics” in the manuscript, since the authors did not perform metabolomic analysis. The authors performed the spectrophotometric analysis of specific metabolites.

Author Response

Reviewer #1:

Line 25: turn “;” to comma. Line 28: turn “relative to” to “compared to”, turn “;” to comma. Line 29: turn “;” to comma. Line 32: add “.”. Line 36: turn “that originated” to “native of”. Line 49: add “one of the most”. Line 36: turn “administering” to “administer”. Line 372: remove "to the"

Answer: Thank you for pointing out the formatting error of this article, and we have corrected it according to your requirements at abstract section and Ln37, 50, 52 and 383.

Line 321-322: they have a function against pathogen and deterrent action against insects. I suggest to rephrase this sentence. Line 368: I suggest to rephrase this point...help in the maintainance of the photosynthetic apparatus...please be careful. The authors did not measure photosynthesis.

Answer: Thank you for your suggestions. We have modified the sentence “Glucosinolates have been reported to function as disease and resistance compounds and as antioxidants in plants” to “Glucosinolates have been reported to function as against pathogen and deterrent action against insects” at Ln331-332, and modified the sentence “help in the maintainance of the photosynthetic apparatus” to “LED light treatment helped to up-regulated the the expression of genes related to photosynthesis” at Ln374-375.

Line 57: it would be important to cite papers on fruits such as this recent work:  Huang, J. Y., Xu, F., & Zhou, W. (2018). Effect of LED irradiation on the ripening and nutritional quality of postharvest banana fruit. Journal of the Science of Food and Agriculture, 98(14), 5486-5493.

Answer: Thank you for your suggestions. We have added two references on fruits at Ln56-59 and 437-441.

Line 68: Add the reference of this paper.

Answer: Thank you for your suggestion. We have added two references of pak-choi published by our team earlier at Ln65 and 69. We didn't add these in the original manuscript because these were not published at the time.

Line 68: please add some more details.

Answer: Thank you for your suggestion. We have added more information about method of the RNA-seq library preparation and sequencing at Ln106-111.

Line 60: The authors cannot use the word metabolomic because they did not use neither a classic untarget metabolomoic approach nor a metabolomic characterization of the plant tissues using HPLC-MS/MS analysis. I suggest to remove the terms "metabolomic analysis" and metabolomic approach". Line 177: turn “metabolomic” to “analysis of specific metabolites”. Line 354: see comments above on this point.

Answer: Thank you for your suggestion. In our previous work, we used an untargeted metabolomic approach to analyse the mechanism of LED treatment maintain postharvest quality of pak-choi (Zhou et al., 2019). Therefore, the main theme of this research is to combine the metabolomics data obtained in the past with the transcriptomic data we just completed analysis, in order to further explore the mechanism of pak-choi by LED irradiation. To make the manuscript clearer, we have cited previous research data, rewrote the abstract section, supplemented the introduction section and revised the results and discussion section. And in the conception of this paper, we refer to the research structure of Hirasawa et al. (2018).

References:

Zhou, F. H.; Zuo, J. H.; Gao, L. P.; Sui, Y.; Wang, Q.; Jiang, A. L.; Shi, J. Y. An untargeted metabolomic approach reveals significant postharvest alterations in vitamin metabolism in response to LED irradiation in pak-choi (Brassica campestris L. ssp. chinensis (L.) Makino var. communis Tsen et Lee). Metabolomics. 2019, 15(12), 155.

Hirasawa, T., Saito, M., Yoshikawa, K., Furusawa, C., & Shmizu, H. Integrated Analysis of the Transcriptome and Metabolome of Corynebacterium glutamicum during Penicillin‐Induced Glutamic Acid Production. Biotechnology journal. 201813(5), 1700612.

Reviewer 2 Report

I think that this manuscript describing an integrated transcriptomic and metabolomic analysis of a low-intensity white LED light treatment on pak-choi is little originality and cannot find scientific significance. First, since the authors quantify only three metabolites, most of the manuscript describes the results of transcription analysis and has not led to an integrated analysis. Second, the definition of post-harvest quality is ambiguous. There are various aspects, whether nutritional value, appearance, moisture content, or consumer demand, and should be defined properly. Third, in comparisons between samples, various light qualities, illuminances, and irradiation times have not been examined, and it is unlikely that they contain scientifically new content.

 Therefore, I think this manuscript should be published in journals in other specialized fields.

Author Response

Ref: biomolecules-634044

Title: Integrated analysis of transcriptomic and metabolomic data reveals the mechanism by which LED light irradiation extends the postharvest quality of pak-choi (Brassica campestris L. ssp. chinensis (L.) Makino var. communis Tsen et Lee).

Point-to-Point Response to comments made by the reviewers

Reviewers' comments:

Reviewer #2:

First, since the authors quantify only three metabolites, most of the manuscript describes the results of transcription analysis and has not led to an integrated analysis.

Answer: Thank you for the comments. In our previous research, it has been found that LED treatment can effectively maintain the quality of post-harvest pak-choi (Zhou et al., 2019) and increase the content of nutrients such as vitamins (Zhou et al., 2020). In order to obtain further information, we supplemented transcriptomics and combined analysis with metabolomics to identify gene-to-metabolite networks involved in the physiological responses of pak -choi to LED treatment. Because our previous manuscript has a detailed analysis of metabolomics, in this study, we focused on the transcriptomics and analyzed the correlation between its results and metabolomics. To make the manuscript clearer, we have cited previous research data, rewrote the abstract section, supplemented the introduction section and revised the results and discussion section. 

Second, the definition of post-harvest quality is ambiguous. There are various aspects, whether nutritional value, appearance, moisture content, or consumer demand, and should be defined properly.

Answer: Thank you for raising this concern. In our previous research, we found that low-intensity (10 μmol m-2 s-1) white LED treatment effectively delayed senescence and maintained the quality of postharvest pak-choi during storage (Fig. 1) and found that the underlying mechanisms included reduced respiration and lower accumulation of MDA, stimulation of antioxidant gene expression and enzyme activity, and the regulation of chlorophyll metabolism gene expression and enzyme activity (Zhou et al., 2020). Then we used metabolomics for further analysis. The result demonstrated that LED irradiation maintaining the quality of pak-choi by regulating several metabolic processes, elevating folate, thiamine, glutathione, riboflavin and total carotenoid content, decreasing the degradation of glucosinolate, and enhancing key synthetic precursors of chlorophyll (Fig.2) (Zhou et al., 2019). The above two manuscripts have systematically research the appearance and nutritional quality of postharvest pak-choi. Therefore, this study used transcriptomics to obtain more information about pak-choi response to LED irradiation from the molecular perspective.

Based on your suggestions, we have clearly stated the purpose of this article in the abstract section at Ln20-22. Additionally, we have supplemented the introduction and results and discussion sections in the revised manuscript, and cited our previous manuscript.

Fig. 1. The effect of different intensities of white LED on the sensory quality of pak-choi.

Fig. 2. The level of (A) thiamine, (B) glutathione, (C) total glucosinolate, (D) riboflavin and (E) total carotenoids in the three sample groups (initial, dark-treated, and light treated). Vertical bars are SD (n = 3). Asterisks indicate significant treatment differences (p < 0.05) after five days of storage.

References:

Zhou, F. H.; Zuo, J. H.; Xu, D. Y.; Gao, L. P.; Wang, Q.; Jiang, A. L. Low intensity white light-emitting diodes (LED) application to delay senescence and maintain quality of postharvest pakchoi (Brassica campestris L. ssp. chinensis (L.) Makino var. communis Tsen et Lee). Scientia Horticulturae. 2020, 262, 109060.

Zhou, F. H.; Zuo, J. H.; Gao, L. P.; Sui, Y.; Wang, Q.; Jiang, A. L.; Shi, J. Y. An untargeted metabolomic approach reveals significant postharvest alterations in vitamin metabolism in response to LED irradiation in pak-choi (Brassica campestris L. ssp. chinensis (L.) Makino var. communis Tsen et Lee). Metabolomics. 2019, 15(12), 155.

Third, in comparisons between samples, various light qualities, illuminances, and irradiation times have not been examined, and it is unlikely that they contain scientifically new content.

Answer: Thank you for your comments and concerns. We have added detailed parameters of the LED to the material and methods section at Ln91-96 and added the type of the light intensity measurement instrument at Ln95-97. In previous reports, we described the parameters and effects of LED on postharvest pak-choi quality in detail (Zhou et al., 2019; Zhou et al., 2020). In this manuscript, we added a detailed description of these two manuscripts at Ln63-69, and cited these two articles at Ln445-451.

Additionally, in the revised manuscript, we have rewritten the abstract and the introduction sections to explain that the purpose of this study is to combine the transcriptomics with metabolomics for joint analysis. Our main focus is on the mechanism of LED treatment to maintain the quality of postharvest pakchoi.

Reviewer 3 Report

1)  the authors do not take into account the influence of the LED on other important and fundamental metabolites for the nutritional quality of Pak-choi, such as total polyphenols and vitamins. It is possible  

2) The experimental plan should be expanded. The authors should detail the method of determination of chlorophylls, because the cited bibliographical reference refers to other plants.

Author Response

Title: Integrated analysis of transcriptomic and metabolomic data reveals the mechanism by which LED light irradiation extends the postharvest quality of pak-choi (Brassica campestris L. ssp. chinensis (L.) Makino var. communis Tsen et Lee).

Point-to-Point Response to comments made by the reviewers

Reviewers' comments:

Reviewer #3:

the authors do not take into account the influence of the LED on other important and fundamental metabolites for the nutritional quality of Pak-choi, such as total polyphenols and vitamins. It is possible

Answer: Thank you for your comments. Before reporting this study, we did a comprehensive quality study of pak-choi and found that LED treatment can increase the antioxidant capacity and regulate chlorophyll metabolism of postharvest pak-choi (Zhou et al., 2020). Additionally, by analyzing untargeted metabolomic data, we found that LED treatment can promote vitamin content of postharvest pak-choi including: folate, thiamine, glutathione, riboflavin and total carotenoid (Zhou et al., 2019). Therefore, in this manuscript, we performed a joint analysis of transcriptomic and metabolomic to further understood that gene-to-metabolite networks involved in the physiological responses of postharvest pak-choi to LED treatment. To make the manuscript clearer, we rewrote the abstract section, added the results of our previous research in the introduction at Ln63-69, cited these two articles at Ln445-451 and revised the introduction and results and discussion sections.

The experimental plan should be expanded. The authors should detail the method of determination of chlorophylls, because the cited bibliographical reference refers to other plants.

Answer: Thank you for suggestions. We have expanded the method of determining chlorophyll content at Ln 172-175 and added the reference (Zhang, L.; Xiao, S.; Chen, Y. J.; Xu, H.; LI, Y. G.; Zhang, Y. W.; Luan, F. S. Ozone sensitivity of four pakchoi cultivars with different leaf colors: physiological and biochemical mechanisms. Photosynthetica. 2016, 55(3), 478-490.) at Ln 482-484. In this reference, the chlorophyll content of pak-choi was determined.